# Botulinum Toxin Type A Injections Impact Hamstring Muscles and Gait Parameters in Children with Flexed Knee Gait

**DOI:** 10.3390/toxins12030145

**Published:** 2020-02-27

**Authors:** Seung Ki Kim, Dong Wook Rha, Eun Sook Park

**Affiliations:** Department of Rehabilitation Medicine, Severance Hospital, Research Institute of Rehabilitation Medicine, Yonsei University College of Medicine, 50 Yonsei-ro, Seodaemun-gu, Seoul 03722, Korea; gsg1230@yuhs.ac (S.K.K.); medicus@yuhs.ac (D.W.R.)

**Keywords:** botulinum toxin type A, cerebral palsy, children, hamstring muscles, injection, knee flexion gait, semimembranosus muscle

## Abstract

The aim of this study was to determine if botulinum toxin type A (BoNT-A) injection into the medial hamstring can improve gait kinematics and muscle-tendon length in spastic cerebral palsy (CP) with a flexed knee gait (FKG). Twenty-nine children with spastic CP (Gross Motor Function Classification System I–III) with FKG were recruited for this prospective study. BoNT-A was injected into the semitendinosus and semimembranosus (SM) muscles under ultrasonography guidance. Assessments included Gross Motor Function Measure (GMFM), Modified Ashworth Scale (MAS), Modified Tardieu Scale (MTS), 3-dimensional computerized gait analysis, calculated SM muscle-tendon length and lengthening velocity during gait using musculoskeletal modeling at baseline, 4 and 16 weeks after the injection. Compared to baseline data, significant improvements in GMFM, MAS, and MTS were demonstrated at weeks 4 and 16, and also a significant increase in maximum knee extension during the stance phase was observed at week 4. In addition, the mean lengthening velocity during the swing phase was increased at week 16 without a change in the SM muscle length. Furthermore, there was a significant increase in anterior pelvic tilt at week 4, compared to baseline data. The significant decrease in hip internal rotation after injection was observed only in children with excessive hip internal rotation at initial contact before injection. BoNT-A injection into hamstrings leads to a significant increase in knee extension and anterior pelvic tilt with an increase in lengthening velocity of SM in spastic CP with FKG.

## 1. Introduction

Flexed knee gait (FKG) is a common gait abnormality among children with spastic cerebral palsy (CP) [1], it substantially limits mobility and typically many children get worse over time and lose their independence in functional mobility [2,3]. The cause of FKG is multifactorial but the shortened and spastic hamstring is considered to be the primary cause of this gait abnormality [4].

There were various studies on the length of hamstring in children with flexed knee gait. Some studies reported that the length of the hamstring was shorter in gait analysis of children with CP [5,6], while others suggested that the length of the hamstring is not shorter and only the rate of change is slower than normal [7].

Botulinum toxin type A (BoNT-A) injection has been widely accepted as a safe and effective intervention to control lower limb spasticity in children with spastic CP [8,9,10]. According to a Cochrane review [11], the toxin injection into the lower limb in children with CP has a significant effect in tone reduction and spasticity reduction. In addition, there have been only several reports showing the changes in the parameters of computerized gait analysis, in which there was a significant improvement of ankle dorsiflexion at initial contact after the toxin injection into calf muscles [11]. However, to the best of our knowledge, the effect of BoNT-A injections into the hamstring on the muscle-tendon length and gait kinematics in children with FKG has been rarely studied [12,13,14]. The significant gain in maximum knee extension (KE) after the toxin injection into hamstring muscles were reported in two studies [13,14]. The increased KE at initial contact (IC) were also noted only in Corry et al.’s report [13]. There has been only one report showing a significant increase in muscle length of semimembranosus (SM) and semitendinosus (ST) after toxin injection into hamstring muscles [12].

Therefore, the aim of this study was to investigate the changes in the gait kinematics and muscle tendon length of semimembranosus after BoNT-A injection into the medial hamstring muscles of children with spastic CP with FKG.

## 2. Results

Twenty-nine ambulatory children with spastic CP (23 bilateral CP, 6 unilateral CP; 19 boys and 10 girls) were recruited for this study. The mean age of the subjects was 7.1 years (±3.0), ranging from 3 to 14 years. The children were at Gross Motor Function Classification System (GMFCS) level I to III (6/12/11) (Table 1).

### 2.1. Clinical and Functional Evaluations

The Gross Motor Function Measure-88 (GMFM-88) and the GMFM-66 scores showed statistically significant improvements after injection. Post hoc analysis demonstrated a significant increase in GMFMs scores at 4 and 16 weeks after injection compared to baseline, and significant increases at 16 weeks after injection compared to 4 weeks after injection. Among GMFM items, there were statistically significant changes in crawling and kneeling, standing and walking, and running and jumping. Post hoc analysis showed statistically significant improvement at 16 weeks after injection compared to baseline and 4 weeks after injection (Table 2).

The Modified Ashworth Scale (MAS) and Modified Tardieu Scale (MTS) scores for the knee flexor were significantly decreased at both 4 and 16 weeks after injection compared to baseline data, but there was no significant difference between 4 and 16 weeks after injection (Table 2).

### 2.2. Gait Analysis Using a Computerized Gait System

Gait analysis was performed for all participants but the data for five children with GMFCS level III could not be analyzed due to their poor performances. Therefore, data for gait parameters were analyzed in 24 children. Positive kinematic data values indicated pelvic anterior tilt, hip flexion, knee flexion (KF) and hip internal rotation (IR), while negative values indicated pelvic posterior tilt, hip extension, KE, and hip external rotation (ER).

Spatiotemporal parameters during walking were not significantly different after injection. The kinematic data of computerized gait analysis indicated that significant increase in maximum KE during the stance phase was demonstrated at 4 weeks after injection, but not at 16 weeks, compared to baseline data. The hip flexion at IC, hip flexion at end swing and hip rotation at IC were significantly different between assessments, but the post hoc analysis did not reveal any statistical significance between assessments. In comparison, there were no significant changes after injection in knee flexion at IC (KFIC), maximum KF in swing phase and KF at the end of swing phase (Table 3).

The participants were divided into two groups based on the degree of hip IR at baseline assessment. The excessive hip IR group, group 1 (*n* = 11) was defined as participants with hip rotation angle at IC > −10° and the rest of the participants (*n* = 13) were grouped into group 2. The hip IR at IC and the mean IR in swing phase were significantly decreased in group 1, but not in group 2 (Figure 1).

#### Curve Analysis Using Statistical Parametric Mapping (SPM) Analysis

The knee angle pattern in the sagittal plane showed significant differences from 46% to 62% of the gait cycle between assessments (*p* = 0.003) (Figure 2A). Post hoc analysis showed a significant increase in KE at 4 weeks after injection compared to baseline evaluation (*p* = 0.001). The pelvic angle in the sagittal plane also showed significant differences from 10% to 21% of the gait cycle between assessments (*p* = 0.038) (Figure 2B). Post hoc analysis showed a significant increase in anterior pelvic tilt at 4 weeks after injection, compared to baseline evaluation (*p* < 0.001). The pelvic angles in the coronal and transverse planes and the hip angles in the sagittal, coronal and transverse planes during walking were not significantly different between baseline, 4 weeks and 16 weeks after injection (Figure 3).

### 2.3. Muscle-Tendon Length

The mean SM lengthening velocity during the swing phase increased significantly at 16 weeks after injection, compared to baseline on post-hoc analysis (*p* = 0.025). However, the SM length at IC, minimum length during stance phase, maximum length during swing phase, length at end of swing and maximum lengthening velocity during swing phase were not significantly different after injection (Table 3).

## 3. Discussion

In this study, the gross motor function steadily increased up to 16 weeks after BoNT-A injection although the tone reduction at 4 weeks post-assessment was maintained until 16 weeks post-assessment. The results are in agreement with findings of previous studies in which BoNT-A intervention has a long term effect on gross motor function with intensive physical therapy after injection, although the effect of muscle tone disappeared [15,16,17,18]. However, the improvement in GMFM did not lead to improvement in spatiotemporal parameters of gait in this study. The participant’s walking ability was somewhat limited in most of the children (GMFCS level II or III) and thus, gross motor functional gain did not translate into spatiotemporal parameters of gait.

In this study, there was a significant change in KE during stance phase, but not in KFIC after injection. The BoNT-A injection into hamstring muscles led to an improvement in maximum KE in the stance phase in the children with a crouch gait [12]. The KFIC is affected by additional factors such as selective motor control and timing of maximum KF, and thus the KFIC did not reveal any relationship with the popliteal angle or KF contracture [2]. In addition, the maximum length of SM was more closely related to KF during single-limb support (KFSLS) than with KFIC [2]. Thus, KFSLS, rather than KFIC, could improve after BoNT injection into hamstring.

The SPM analysis allows kinematic and kinetic waveforms to be analyzed as a whole or particular gait phase [19]. To the best of our knowledge, there is only one report that has shown the changes of gait kinematic data using SPM analysis after BoNT-A injection; the results of the study indicated that there was a significant increase in KE during terminal stance that might be related to the hamstring injection [19]. The study was a retrospective study and various muscles of the lower extremity from the hip to ankle joint muscles had were injected with BoNT-A, although most of the injections were into hamstring and calf muscles. In this study, a significant increase in KE during stance phase was found on both statistical approach methods. The SPM analyses also showed that anterior pelvic tilt was significantly increased after the hamstring injection. The medial hamstring muscle is a two-joint muscle that acts as both a knee flexor and a hip extensor. The complex interactions of hamstring muscles on the pelvis were shown in previous studies [20]. Hamstring length was significantly related with pelvis tilt in both CP children and normal control children [21]. The dynamic spasticity of the hamstring muscle might be act as a contributor of posterior pelvis tilt during gait [20]. On the other hand, concurrent spasticity in other lower limb muscles is common in children with CP. Corry et al. reported an increased anterior pelvic tilt after injecting the toxin into the hamstring if the iliopsoas was left untreated [13]. In this study, the anterior tilted pelvis before injection suggests the presence of concurrent hip flexor spasticity. These findings indicate that a BoNT-A injection into the hamstring muscle may lead to anterior pelvic tilt if the iliopsoas is left untreated, which is concerning, especially for cases with concurrent hip flexor spasticity.

Hamstring lengthening surgery could lead to a significant decrease in hip IR regardless of the status of hip internal or external rotation preoperatively [22]. In this study, hip IR was decreased only in the group with excessive hip IR before the intervention. These findings suggest that the BoNT-A injection into the medial hamstring may be helpful to reduce toe-in-gait in these children. However, the sample size of subgroups in the present study was small, therefore additional studies that address kinematic changes at the hip joint are needed with a large sample size.

Several studies on hamstring length in children with FKG have been reported. Some studies reported that the length of the hamstring was shorter in gait analysis of children with CP [5,6], while others suggested that the length of the hamstring is not shorter and only the rate of change is slower than normal [7]. BoNT-A influences the dynamic component of the contracture and makes relatively small changes in absolute muscle length [12]. The increase in SM lengthening velocity without a significant change in SM length in the present study appears to be related to the action characteristics of BoNT-A. The lengthening velocity during the swing phase is associated with other factors, including co-contraction or selective motor control, as well as dynamic spasticity and muscle shortening [20]. In the present study, we observed a significant improvement in SM lengthening velocity without significant changes in KFSLS at 16 weeks after injection. Considering the improvements in GMFM at post 16-week assessment compared with the post 4-week assessment, the improvement in SM lengthening velocity may likely result from the influences of the other factors, such as selective motor control and knee extensor muscle strength, along with dynamic spasticity or shortening of a muscle.

The lack of a control group is a limitation of the present study. Gross motor function can be improved with intensive therapy without BoNT-A injection. Therefore, it is still under question whether BoNT-A injection combined with physical therapy is conducive to greater improvement compared to physical therapy alone. However, the changes in knee kinematics and SM lengthening velocity without significant changes in other spatiotemporal parameters in the present study seem to be mostly caused by the BoNT-A injection into the hamstring of children with a flexed knee gait. Further study with a control group is needed to delineate the additional effects of BoNT-A injection into hamstrings on these parameters. Additionally, the musculoskeletal model used to estimate SM length in this study did not reflect the bony deformities or changed muscle physiology of children with CP. Further studies with a new musculoskeletal model that reflect individual characteristics are needed. The small sample size is another limitation of this study. It is possible that other kinematic parameters that were not determined to have significant impacts may be associated with significant changes after injection in a large sample size.

## 4. Conclusions

In spastic CP with FKG, a BoNT-A injection into the hamstring combined with intensive physical therapy led to significant improvements in GMFM, KE during the stance phase and mean SM lengthening velocity without a change in SM length. In addition, the injection into the medial hamstring led to a significant decrease in hip IR in the cases with excessive hip IR before injection. Using SPM analysis, we found additional significant changes in the anterior pelvic tilt. These findings indicate that BoNT-A injection into the hamstring may lead to further anterior pelvic tilt which could be concerning for children with concurrent hip flexor spasticity if the iliopsoas is left untreated.

## 5. Materials and Methods

This prospective intervention study was conducted in a university-affiliated, tertiary-care hospital between November 2016 and September 2018.

### 5.1. Participants

This study recruited children with spastic CP who met the following inclusion and exclusion criteria (see Table 1). The inclusion criteria were as follows: (1) children with spastic CP between 2 and 18 years of age, (2) the ability to walk independently without assistive devices for at least 6 m, and (3) MAS at knee flexors ≥ 1+ with R1-R2 angle of MTS ≥ 15°. Exclusion criteria were as follows: (1) chemodenervation therapy or serial lower extremity casting within 6 months, (2) previous selective rhizotomy, orthopedic surgery, or an intrathecal baclofen pump, (3) history of allergy to the toxin, and (4) change in oral medications and dosing that might influence muscle tone within 30 days. In the case of bilateral CP, only the more-involved limb, based on R1-R2 angle of MTS, was analyzed for this study. Ethical approval was granted by the Institutional Review Board and Ethics Committee of the Severance Hospital (#4-2016-0265). Informed consent was obtained from the primary caregiver and/or the participant along with written minor assent according to the rules of the IRB of our hospital.

#### Sample Size

The sample size was calculated using G*power software (version 3.1) [23]. There was no previous study to estimate the effect size, and the medium effect size 0.25 proposed by Cohen was used [24]. When a repeated measures ANOVA with a significance level of 5% (α error) and a test power of 80% were applied, the sample size was calculated as 28 [25]. Assuming a 10% loss to follow-up, 31 patients were needed for this study.

### 5.2. Intervention

We applied topical local anesthetic lidocaine cream, Kolmar Korea Co. Ltd., Sejong-si, Korea) at the injection site 20 min before the BoNT-A injection. Reconstituted vials containing 500 units of abobotulinum toxin A (Dysport^®^, Ipsen Ltd., Slough, UK) with 2.5 mL of normal saline to provide a solution that contained 200 units/mL were prepared for injection. BoNT-A was injected into both the ST and SM muscles under the guidance of ultrasonography. The injection was directed according to the intramuscular nerve distribution of the hamstring muscle [26]. Thus, the BoNT-A was injected at about distal two-thirds point from the proximal insertion of SM and ST (Figure 4). The dose of the toxin for each muscle ranged from 5.0 to 7.5 units/kg depending on the severity of spasticity (mean ± SD = 6.0 ± 1.1 U/kg). Total doses per each muscle did not exceed a maximum of 250 units, and total maximal dose per child was set 1000 units. Most of the participants had intensive physical therapy including stretching, strengthening and gait training for 16 weeks after injection (at least 4 sessions per week).

For each child, Gross Motor Function Classification System-Expanded and Revised (GMFCS-E&R), Gross Motor Function Measure (GMFM), clinical spasticity evaluation (MAS, MTS), and computerized gait analysis were performed at the time points before BoNT-A injections, 4 weeks and also 16 weeks after injection.

#### 5.2.1. Functional Evaluation

The gross motor function was assessed using the GMFCS-E&R and GMFM-88 and -66. The GMFCS-E&R has five levels based on self-initiated movement for children with CP from I (most able) to V (least able) [27]. The GMFM is a functional outcome tool to measure changes in gross motor function of children with CP. The GMFM-88 is the original 88-item measure, whose items are made up of five dimensions (lying and rolling; sitting; crawling and kneeling; standing; and walking, running and jumping) [28]. The GMFM-66 is a 66 item subset of the original 88 items and unidimensional scale providing interval scaling [29].

#### 5.2.2. Spasticity Assessment

We assessed the spasticity of the knee flexor with the MAS and MTS in the supine position. The MAS is a 6-point ordinal scale from 0 to 4 used to measure muscle tone [30]. For statistical analysis, a MAS grade of 1 was considered as 1, while a grade of 1(+) was regarded as 2, up to a score of 4, which was regarded as a 5. The MTS was a method of measuring the popliteal angle at two levels with manual goniometry after slow and fast stretching of the knee joint and those angles were referred to R2 and R1 angles respectively [31]. The R1 angle was the point in the range of motion where a catch was first felt during a quick, passive extension of the knee joint, while the R2 angle the popliteal angle measured at the end of the motion. The difference between the two angles (R1-R2) represents the dynamic elements of spasticity [32]. All clinical evaluations were recorded by an experienced physiatrist (S.K.K.).

#### 5.2.3. Gait Analysis Using a Computerized Gait System

We conducted gait analysis using a computerized three-dimensional motion analysis system (VICON MX-T10 Motion Analysis System, Oxford Metrics Inc., Oxford, UK) for measuring kinematic data during the gait cycle. According to Helen Hayes marker, a trained investigator who had 20 years of clinical experience in gait analysis, we attached 16 passive reflex markers to subjects. While the child walked barefoot on an 8-m pathway, six digital videos on the front, rear and sides were recorded simultaneously. Data were collected at a sampling rate of 100 Hz from three trials at self-selected walking speed. We instructed each patient to look straight ahead and walk as naturally as possible. All data were captured based on the VICON Plug-in-Gait model [33]. Then, joint kinematics were calculated based on an average of three representative trials using NEXUS software version 1.8.5.

#### 5.2.4. Muscle-Tendon Length

We measured the muscle-tendon length and velocity of the SM in each subject. Since the length changes of SM and ST were known to be similar during walking in previous studies [6,34], this study analyzed only SM. We used Lower Limb Extremity Model 2010 [35], a 3D computer model of the lower limb, to estimate the muscle-tendon length. We used OpenSim [36], an open-source biomechanics simulation application, to conduct an inverse kinematic analysis of motion capture data with this model. In order to measure the changes of muscle-tendon length during walking, we used the least-squares formulation to calculate a set of desired joint angles for tracking based on marker trajectories from gait analysis. Muscle-tendon lengths were normalized based on the lengths when the hip, knee, and ankle were in the anatomic position, with all joint angles set at zero [7]. Three steps from a multi-step trial were averaged to analyze the muscle-tendon length during the gait cycle. We estimated the muscle-tendon lengthening velocity by using the “approximate derivatives with diff” function in Matlab software (version R2016a, 2016; Mathworks Inc, Natick, MA, USA) to computing the numerical derivative of the muscle-tendon length data with respect to time [5].

### 5.3. Statistical Analysis

We performed SPSS Statistics for Windows (version 25.0, 2017; IBM Corp., Armonk, NY, USA) for statistical analysis. A one-way repeated-measures analysis of variance (ANOVA) was performed to compare the changes from baseline at 4 weeks and 16 weeks after injection. The post hoc test used the Bonferroni method to further analyze the time factor. The level of significance was set as *p* < 0.05.

In addition, kinematic data were analyzed using one-dimensional SPM, for curve analysis, which allows hypothesis testing on kinematic waveforms without the need for reducing a priori data [37]. SPM one-way-ANOVA was performed to examine the statistical significance of the change in kinematic data for each time point. SPM *t*-tests were performed to compare the kinematic angle of each joint to the respective kinematic angle (α = 0.05). For each SPM ANOVA or t-test, Computing the conventional univariate F- or t-statistic at each point of the gait waveform created a SPM, referred as SPM(F) or SPM(t) [38]. Next, the Random Field Theory was used to estimate the critical threshold above which only 5% (α = 0.05) of smooth random curves would be expected to traverse [39]. If the SPM(F) curve exceeded the critical threshold, post-hoc SPM(t)was computed for between-group comparisons. If at any time, an SPM(t) curve exceeded the critical threshold, a supra-threshold cluster was created that represents a significant difference between two joint angles in a specific section of the gait cycle. We performed a Bonferroni correction to adjust α for multiple post-hoc comparisons. When testing equally smooth random data for each supra-threshold cluster, we calculated the probability of finding a cluster with similar proportions [39]. All analyses were performed using open-source SPM1D code (vM.01.0003; www.spm1D.org) in Matlab software.

## Figures and Tables

**Figure 1 toxins-12-00145-f001:**
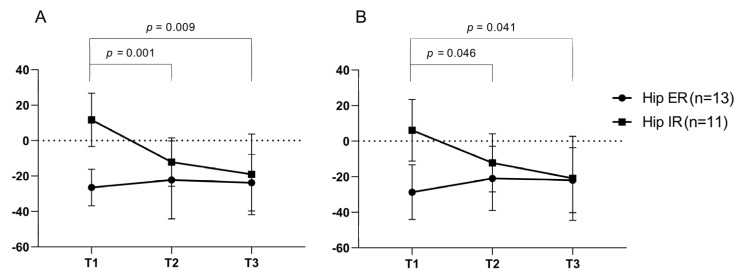
Hip internal rotation angle by hip rotation subgroup. Hip internal rotation group (Hip rotation angle at initial contact > −10°, *n* = 11) showed a significant decreased in internal rotation at initial contact (**A**) and mean internal rotation in swing phase (**B**) over time after the injection. However, the hip external rotation group (hip rotation angle at initial contact < −10°, *n* = 13) did not demonstrate significant changes over time.

**Figure 2 toxins-12-00145-f002:**
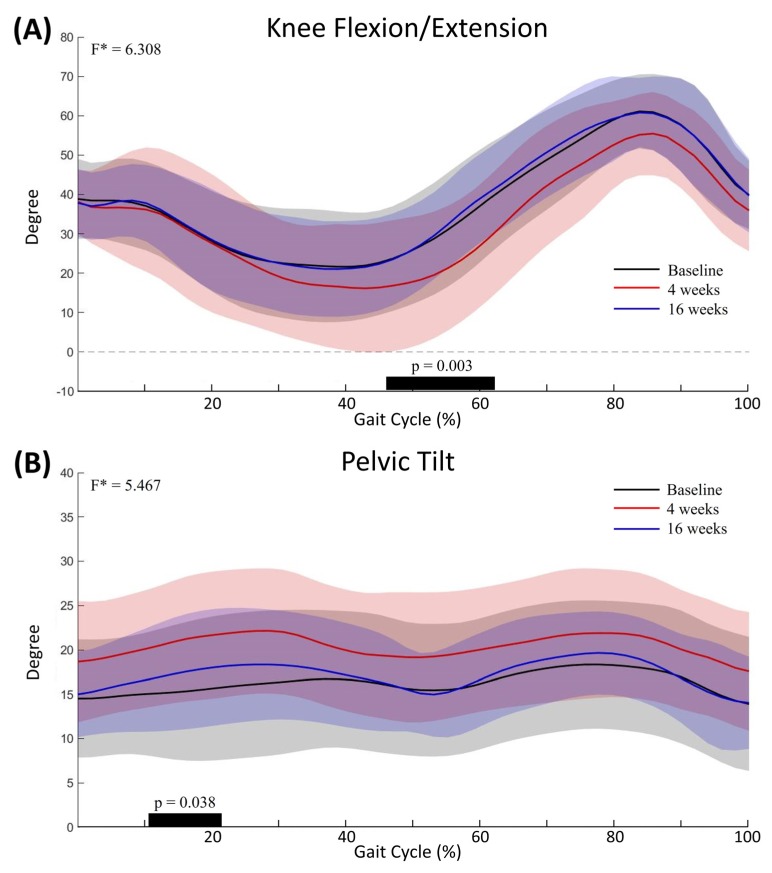
Mean knee (**A**) and pelvic (**B**) angles in the sagittal plane in 24 participants. Graphs show the mean kinematic angle during gait cycle for baseline (black), 4 (red) and 16 (blue) weeks after injection. Shaded areas indicated ± 1 SD. Thick black bar represents significant differences between assessments. All curves were normalized over the gait cycle (%).

**Figure 3 toxins-12-00145-f003:**
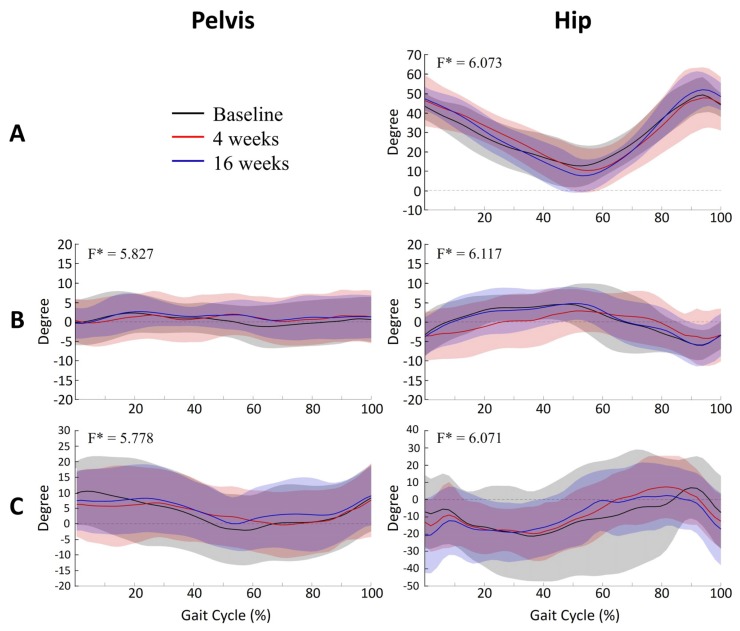
Mean kinematic angle during the gait cycle at the level of pelvic and hip joints in 24 participants. Each graph shows the respective mean kinematic angle during the gait cycles. The pelvis and hip angles in all planes did not reveal any significant differences between assessments, with the exception of the pelvic angle in the sagittal plane. (**A**), sagittal plane; (**B**), coronal plane; (**C**), transverse plane.

**Figure 4 toxins-12-00145-f004:**
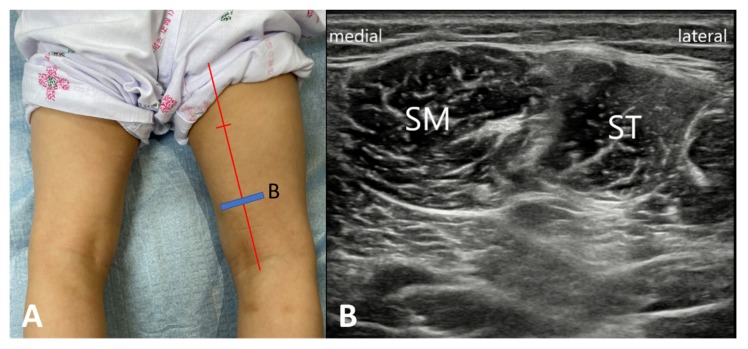
The botulinum toxin A was injected at about distal two-thirds point of the muscle length under the guidance of ultrasonography. (**A**) The blue line indicated probe positioning. (**B**) The transvers view of ultrasound showed semimembranosus (SM) and semitendinosus (ST) muscles.5.3. Assessment.

**Table 1 toxins-12-00145-t001:** Participant Characteristics.

Characteristic	
Age (mean ± SD), year	7.1 ± 3.0
Sex (Male/Female), n (%)	19 (65.5%)/10 (34.5%)
CP type (Bilateral/Unilateral), n (%)	23 (79.3%)/6 (20.7%)
GMFCS level (I/II/III), n (%)	6 (20.7%)/12 (41.4%)/11 (37.9%)
Exam side (right / left), n (%)	16 (55.2%)/13 (44.8%)

CP, cerebral palsy; GMFCS, gross motor function classification system.

**Table 2 toxins-12-00145-t002:** Clinical and Functional Assessment.

Parameters	T1 (before injection)		T2 (4 weeks after injection)		T3 (16 weeks after injection)		ANOVA *p*-value	Post hoc analysis		
	Mean	SD	Mean	SD	Mean	SD		T1 vs. T2	T1 vs. T3	T2 vs. T3
**GMFM-88 (%)**										
A. Lying and Rolling	100	0	100	0	100	0	1.000			
B. Sitting	99.48	1.35	99.6	1.06	99.71	0.64	0.228			
C. Crawling and Kneeling	91.38	8.36	92.04	6.74	93.84	6.12	0.007	1.000	0.024	0.004
D. Standing	59.34	24.17	60.83	23.38	64.55	23.08	0.000	0.295	0.000	0.008
E. Walking, Running and Jumping	42.72	25.83	43.58	26.08	46.88	27.05	0.001	0.373	0.004	0.001
Total score	78.56	11.03	79.24	10.67	81.17	10.96	0.000	0.011	0.000	0.001
GMFM-66	61.62	9.47	62.12	9.59	63.43	9.81	0.000	0.016	0.000	0.001
**Knee flexor (degrees)**										
MAS	2.66	0.67	1.83	0.54	1.97	0.57	0.000	0.000	0.000	0.359
R1 of MTS	53.79	8.83	42.07	7.14	41.9	7.61	0.000	0.000	0.000	1.000
R2 of MTS	32.41	10.91	26.21	6.5	27.24	6.21	0.001	0.000	0.023	0.891
R1-R2 of MTS	21.38	11.87	15.86	7.68	14.66	6.26	0.000	0.001	0.003	1.000

GMFM, gross motor function measure; MAS, modified Ashworth scale; MTS, modified Tardieu scale.

**Table 3 toxins-12-00145-t003:** Changes in Spatiotemporal, Kinematic, and Muscle-Tendon Length Gait Analysis Parameters.

Parameters	T1 (before injection)		T2 (4 weeks after injection)		T3 (16 weeks after injection)		ANOVA *p*-value	Post hoc analysis		
	Mean	SD	Mean	SD	Mean	SD		T1 vs. T2	T1 vs. T3	T2 vs. T3
**Spatiotemporal**										
Cadence (step/min)	107.53	35.49	102.03	47.94	113.63	36.30	0.208			
Walking Speed (cm/s)	55.50	27.01	55.46	35.97	65.29	26.82	0.123			
Step Length (cm)	29.33	11.90	30.21	12.43	32.54	10.46	0.363			
Step Time (s)	0.65	0.36	0.83	0.66	0.68	0.66	0.314			
Step Width (cm)	12.10	4.21	14.06	5.44	12.24	5.31	0.111			
**Pelvis (degrees)**										
Pelvic tilt at IC	16.00	7.22	18.94	7.74	16.76	6.53	0.097			
Max pelvic tilt in ST	21.15	7.96	23.74	7.62	22.65	6.31	0.170			
Pelvic tilt at end swing	15.44	8.24	17.64	7.85	15.55	6.49	0.271			
Mean pelvic tilt in ST	17.31	7.52	20.39	7.54	18.35	5.87	0.066			
**Hip (degrees)**										
Hip flexion at IC	44.29	9.26	48.16	9.47	48.39	7.76	0.036	0.078	0.106	1.000
Max hip extension in terminal ST	9.33	8.87	6.15	10.48	6.78	7.84	0.190			
Max hip flexion in SW	51.14	11.37	53.72	10.25	55.21	8.70	0.085			
Hip flexion at end swing	43.50	10.18	46.97	9.61	48.31	7.64	0.034	0.239	0.064	1.000
**Knee (degrees)**										
Knee flexion at IC	37.70	10.67	34.94	9.48	36.67	8.54	0.537			
Max knee extension in ST	19.26	12.65	12.19	14.92	18.13	11.61	0.001	0.003	1.000	0.030
Max knee flexion in SW	64.81	9.80	61.45	7.21	65.05	10.06	0.156			
Knee flexion at end swing	36.34	10.46	33.91	9.95	36.86	10.31	0.374			
**Hip (degrees)**										
Hip rotation at IC	−8.96	23.07	−17.56	18.96	−21.62	19.08	0.042	0.290	0.084	1.000
Max hip internal rotation in ST	10.47	24.39	5.87	20.64	8.71	20.37	0.694			
Max hip external rotation in ST	−24.59	24.92	−28.33	17.10	−29.46	16.36	0.522			
Mean hip rotation in ST	−11.57	23.41	−13.90	17.38	−11.90	18.24	0.861			
Max hip internal rotation in SW	15.33	25.15	10.22	19.68	11.57	19.87	0.599			
Max hip external rotation in SW	−19.34	25.03	−19.12	16.39	−23.15	20.15	0.663			
Mean hip rotation in SW	−12.72	23.82	−16.94	17.48	−21.47	20.41	0.249			
Hip rotation at end swing	1.61	22.67	−0.81	18.09	−3.07	16.43	0.616			
**Normalized muscle lengths during gait**										
SM at IC	1.03	0.04	1.04	0.04	1.03	0.05	0.409			
Min SM in ST	0.92	0.04	0.93	0.04	0.91	0.03	0.210			
Max SM in SW	1.05	0.05	1.06	0.04	1.06	0.04	0.434			
SM at end swing	1.00	0.18	1.01	0.17	1.04	0.05	0.294			
Mean SM lengthening velocity in SW	0.31	0.15	0.37	0.18	0.41	0.17	0.017	0.256	0.025	0.737
Max SM lengthening velocity in SW	0.54	0.18	0.61	0.19	0.64	0.17	0.080			

GC, gait cycle; IC, initial contact; Max, maximum; Min, minimum; SM, semimembranosus; ST, stance phase; SW, swing phase. Positive values indicate pelvic anterior tilt, hip flexion, knee flexion, and hip internal rotation while negative values indicate pelvic posterior tilt, hip extension, knee extension, and hip external rotation. Muscle-tendon unit lengths normalized relative to muscle length in the anatomic position.

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
