# Peer review of "Botulinum Toxin Type A Injections Impact Hamstring Muscles and Gait Parameters in Children with Flexed Knee Gait"

_toxins, 2020, doi:10.3390/toxins12030145_

Round 1

Reviewer 1 Report

This article deals with impact of Botulinum toxin type A given in hamstring muscles and its influence on gait in children with CP. The topic is important because improvement of walking influences also several functions and development of CP children. The manuscript is well written. Some comments:

In abstract the authors claim that it was demonstrated significant increase of anterior pelvic tilt and significant decrease in hip internal rotation. Obviously, the authors refer to figure 1 and 2 . It remains unclear how many participants have been included in those Figures. I suggest that n will be added to Figures. In the current version it remains uncertain how representative those findings are. It should be more clearly discussed role of hamstrings in walking ability. Have authors some hypotheses about how treatment of other muscles in addition could result with better results? In Intervention the range of total doses to each muscle should be presented

Author Response

Answer) According to your suggestions, the number of patients are added in figure 1 and 2. In addition, the role of hamstring muscles on gait were more clearly discussed in terms of the results of our study. In terms of dose, we revised the manuscript.

Reviewer 2 Report

This is a nicely performed study with sientific soundness and relevant outcome parameters. However there is no control group as it is a "before-after" design. Where there ethical considerations, or why died you choose an uncontrolled study design? 

Author Response

Answer) We totally agreed with your opinion. There are some reasons for not having control group. Based on our previous studies, the parents or child did not visit to clinic for simple check up without   any financial merit including any free expensive intervention. In addition, computerized gait analysis cost a fortune in our country and thus we could not cover the cost of computerized gait analysis for the control group with our limited budget for this study. Therefore, we have to set this study as before and after design with many clinical and objective measures. We’d like to investigate the changes of the injection on kinematic and muscle length after the injection. Please understand our situation.

Reviewer 3 Report

The authors studied the effect of BoNT-A injections into the hamstring on the muscle-tendon length and gait kinematics in children with FKG. They showed that compared to baseline data, significant improvements in GMFM, MAS, and MTS were demonstrated at weeks 4 & 16. Also a significant increase in maximum knee extension during the stance phase was observed at week 4. In addition, the mean lengthening velocity during the swing phase was increased at week 16 without a change in the SM muscle length. There was a significant increase in anterior pelvic tilt at week 4, compared to  baseline data. The significant decrease in hip internal rotation after injection was observed only in  children with excessive hip internal rotation at initial contact before injection. BoNT-A injection into  hamstrings leads to a significant increase in knee extension and anterior pelvic tilt with an increase in lengthening velocity of SM in spastic CP with FKG.

In Principle, this study is interesting, but proposed few findings concerning the treatment of flexed knee gait in children after CP. However, the manuscript should be dramatically improve before publishing is possible.

Important general questions are:

Why did you include no control group in your study? I think this is a very unappropriate designed. Please add your comments to manuscript.

Why did you examine so few patients compare to other studies?

Why did you study the effect of BoNT-A only in a so short period? Unlu et al. (2010) for example did it until 6 months; Scholtes et al. 2006 during 48 weeks; Fattal-Valevski et al. 2008 during 2 years.

Please add to manuscript List of Abbreviations.

Keywords: Please add also to keywords: injection; children; semitendinosus muscle; semimembranosus muscle  

Introduction

Generally: the Introduction is too short and should be rewritten. Especially:

Line 35: please add references at the end of the sentence.

Lines 34-38: please describe in your introduction very exactly all previous studies and add more appropriate references. 

Lines 39-41 you noted that: "Therefore, the aim of this study was to into medial hamstring muscles in  children with spastic CP with FKG." Why did  only investigated the changes in gait kinematics and muscle  tendon length of semimembranosus after BoNT-A injection? Why not also the changes in gait kinematics and muscle  tendon length of semitendinosus muscle after BoNT-A injection? You injected both muscles - but never come back to the results in semitendinosus muscle. Please explain and add to introduction.

Results

Line 45: why do you say, that childrens´ age was ranging from 3 to 14 years? In Materials and methods you say that age was 2-18 years? Please explain.

Line 45: please write out GMFCS and add reference about this classification system.

Line 48: please write out both and add references about GMFM-88 and GMFM-66.

Lines 49-51: please add the exactly p for significant results after each time point:  Post hoc analysis demonstrated the significant increase in GMFMs scores at 4 weeks and 16 weeks after injection compared to baseline, and significant increases at 16 weeks after injection compared to 4 weeks after injection.

Lines 66-70: you say: "The kinematic data of computerized gait analysis indicated that significant increase in maximum KE during the stance phase was demonstrated at 4 weeks after injection, but not at 16 weeks, compared to baseline data. The hip flexion at initial contact (IC), hip flexion at end swing and hip rotation at IC were significantly different between assessments, but the post hoc analysis did not reveal any statistical significance between assessments." - please add the exactly p for significant results after each time point.

Lines 95-96: Figure 2: what do you mean with (deg)? Please write out and change in Figure.

Lines 100-101: Figure 3.: Figure is too small, please do it little bit bigger. What do you mean with (deg)? Please write out and change in Figure.

Lines 107-110: please add the exactly p for significant results after each time point.

Conclusions:

Line 175: You say: "In spastic CP with FKG, a BoNT-A injection into the hamstring combined with intensive physical  therapy led to significant improvements in GMFM." -  Did you combine BoNT-A injection with intensive physical therapy? Simultaneously you write in the discussion, line 167: "Therefore, the question of whether  BoNT-A injection combined with physical therapy has greater improvement compared to physical therapy only is still under question." Please intensely comment on that facts.

Materials and methods:

Line 186-187: Please add to manuscript in Part "participants": Children with spastic CP (n=?), f=?, m=?, mean age? Or please note (see Table 1).

Line 194: please add references to MTS score.

Line 195: please add hospital name.

Line 199: local anesthetic (lidocaine creme): please add the company, city and country for this medicament, the same as for BoNT-A.  

Lines 202-203: You say: BoNT-A was injected into both the semitendinosus and semimembranosus  (SM) muscles under the guidance of ultrasonography (US). - Please describe very exactly the injection points of BoNT-A into the  semitendinosus and semimembranosus muscles and add to manuscript also an addition Figure that exactly show the injection points into the both muscles.

Line 214: Lines 208-210: please write out and add references after: GMFCS-E&R. please add references after: GMFM

Line 213: please add references  for MAS.

Line 230: please add references at the end of the sentence "We captured all data based on the VICON Plug-in-Gait model."

Line 234: why did you measured only the muscle-tendon length and velocity of the SM in each subject? What about the semitendinosus muscle? Please make a final statement, that the although you injected the semitendinosus muscle this has no remarkable outcome in the parameter measured in the study.

Lines 243-244: please add references at the end of the sentence "We estimated muscle-tendon lengthening velocity by computing the numerical derivative of the muscle-tendon length data with respect to time."

Line 250: please write out SPM.

Lines 246 and 265: please note the publishing year for each software.

Author Response

Important general questions are:

  1. Why did you include no control group in your study? I think this is a very unappropriate designed. Please add your comments to manuscript.

Answer) We totally agreed with your opinion. There are some reasons for not having control group. Based on our previous studies, the parents or child did not visit to clinic for simple check up without   any financial merit including any free expensive intervention. In addition, computerized gait analysis cost a fortune in our country and thus we could not cover the cost of computerized gait analysis for the control group with our limited budget for this study. Therefore, we have to set this study as before and after design with many clinical and objective measures. We’d like to investigate the changes of the injection on kinematic and muscle length after the injection. Please understand our situation.

  1. Why did you examine so few patients compare to other studies?

Answer) Previous studies on gait kinetics or muscle length after botulinum toxin injection into hamstring muscles reported a significant improvement in kinematics at knee joint (Corry et al’s study; n=10, Papadonikolakis et al’s study; n=14) and also a significant improvement in the muscle length (Tompson et al’s study; n= 10) [1-3]. But, there was no previous study to provide us the for this study, and thus,  the medium effect size was set as 0.25 for the calculation of sample size as Cohen suggested [4]. The sample size of this study was calculated using G-power software (version 3.1) (entered ANOVA: Repeated measures, within factors, effect size 0.25, α error probability 0.05, power 0.8, number of groups 1, number of measurements 3, correlation among repeated measures 0.5, and nonsphericity correction ε 1) [5,6]. As a result, the total sample size was calculated as 28. Assuming a 10 % loss to follow up, we recruited 31 participants, but 2 participants had withdrawn from the study throughout the course. As a result, the data of twenty nine participants were included for analysis of this study. The calculation of sample size was added in the manuscript.

  1. Why did you study the effect of BoNT-A only in a so short period? Unlu et al. (2010) for example did it until 6 months; Scholtes et al. 2006 during 48 weeks; Fattal-Valevski et al. 2008 during 2 years.

Answer) There are a lot of factors relating with long term effects of botulinum toxin injection such as child’s cognition, the socioeconomic factors and participation in active physical activities etc. In addition, the effect of BoNT-A injection lasts for 3 months and thus the interval between serial injections ranged from three to six months [7,8]. The goal of this study is to delineate the changes in gait parameters and muscle length resulted from the toxin injection and thus the follow up assessments were done at short term and medium term, not long term.

  1. Please add to manuscript List of Abbreviations.

Answer) As your comment, we added the list of abbreviations to the manuscript.

Keywords

  1. Please add also to keywords: injection; children; semitendinosus muscle; semimembranosus muscle

Answer) As your comment, we added these keywords.

Introduction

Generally: the Introduction is too short and should be rewritten. Especially:

  1. Line 35: please add references at the end of the sentence.

Answer) According to your suggestions, we revised the introduction section.

  1. Lines 34-38: please describe in your introduction very exactly all previous studies and add more appropriate references.

Answer) According to your suggestions, we revised the introduction section.

  1. Lines 39-41 you noted that: "Therefore, the aim of this study was to into medial hamstring muscles in children with spastic CP with FKG." Why did only investigated the changes in gait kinematics and muscle tendon length of semimembranosus after BoNT-A injection? Why not also the changes in gait kinematics and muscle tendon length of semitendinosus muscle after BoNT-A injection? You injected both muscles - but never come back to the results in semitendinosus muscle. Please explain and add to introduction.

Answer) Since the length changes of SM and ST were known to be similar during walking in previous studies [9,10], and only SM was analyzed in this study. According to your comment, we added this in Materials and Methods.

Results

  1. Line 45: why do you say, that childrens´ age was ranging from 3 to 14 years? In Materials and methods you say that age was 2-18 years? Please explain.

Answer) The age of 2 to 18 years was described as one item of inclusion criteria. This was described in the materials and method section. On the other hand, the age of participants who completed the whole procedures for this study, were ranged from 3 to 14 years for this study. This was described in result section.

  1. Line 45: please write out GMFCS and add reference about this classification system.

Answer) As you suggest, we revised the manuscript.

  1. Line 48: please write out both and add references about GMFM-88 and GMFM-66.

Answer) As you suggest, we revised the manuscript.

  1. Lines 49-51: please add the exactly p for significant results after each time point: Post hoc analysis demonstrated the significant increase in GMFMs scores at 4 weeks and 16 weeks after injection compared to baseline, and significant increases at 16 weeks after injection compared to 4 weeks after injection.

Answer) According to your comments, exact p-values were shown in the tables.

  1. Lines 66-70: you say: "The kinematic data of computerized gait analysis indicated that significant increase in maximum KE during the stance phase was demonstrated at 4 weeks after injection, but not at 16 weeks, compared to baseline data. The hip flexion at initial contact (IC), hip flexion at end swing and hip rotation at IC were significantly different between assessments, but the post hoc analysis did not reveal any statistical significance between assessments." - please add the exactly p for significant results after each time point.

Answer) According to your comments, exact p-values were shown in the tables.

  1. Lines 95-96: Figure 2: what do you mean with (deg)? Please write out and change in Figure.

Answer) As you suggest, figure 2 was revised.

  1. Lines 100-101: Figure 3.: Figure is too small, please do it little bit bigger. What do you mean with (deg)? Please write out and change in Figure.

Answer) According to your comment, the figure.3 was revised

  1. Lines 107-110: please add the exactly p for significant results after each time point.

Answer) As you suggest, exactly p-values were added.

Conclusions:

  1. Line 175: You say: "In spastic CP with FKG, a BoNT-A injection into the hamstring combined with intensive physical therapy led to significant improvements in GMFM." - Did you combine BoNT-A injection with intensive physical therapy? Simultaneously you write in the discussion, line 167: "Therefore, the question of whether BoNT-A injection combined with physical therapy has greater improvement compared to physical therapy only is still under question." Please intensely comment on that facts.

Answer) As we mentioned in discussion, gross motor function can be improved with intensive physical therapy. The improvement in GMFM after hamstring injection in our study may be resulted from both the toxin injection and intensive physical therapy. Many previous studies described the y the key role of intensive physical therapy for functional gain after the botulinum toxin injection in the children with CP. Therefore, we have to conclude that both the toxin injection and physical therapy might lead to the changes shown in this study.

Materials and methods:

  1. Line 186-187: Please add to manuscript in Part "participants": Children with spastic CP (n=?), f=?, m=?, mean age? Or please note (see Table 1).

Answer) As you suggest, we revised the manuscript and table 1.

  1. Line 194: please add references to MTS score.

Answer) MTS score means R2-R1 angle of MTS. The reference was added.

  1. Line 195: please add hospital name.

Answer) The hospital name was added in the manuscript.

  1. Line 199: local anesthetic (lidocaine creme): please add the company, city and country for this medicament, the same as for BoNT-A.

Answer) According to your comment, the manuscript was revised.

  1. Lines 202-203: You say: BoNT-A was injected into both the semitendinosus and semimembranosus (SM) muscles under the guidance of ultrasonography (US). - Please describe very exactly the injection points of BoNT-A into the semitendinosus and semimembranosus muscles and add to manuscript also an addition Figure that exactly show the injection points into the both muscles.

Answer) As you suggest, the manuscript was revised and the figure was added

  1. Line 214: Lines 208-210: please write out and add references after: GMFCS-E&R. please add references after: GMFM

Answer) The description of GMFCS-E&R and GMFM was added

  1. Line 213: please add references for MAS.

Answer) The reference was added.

  1. Line 230: please add references at the end of the sentence "We captured all data based on the VICON Plug-in-Gait model."

Answer) The reference was added

  1. Line 234: why did you measured only the muscle-tendon length and velocity of the SM in each subject? What about the semitendinosus muscle? Please make a final statement, that the although you injected the semitendinosus muscle this has no remarkable outcome in the parameter measured in the study.

Answer) Since the length changes of SM and ST were known to be similar during walking in previous studies [9,10], only SM length was analyzed for this study.

  1. Lines 243-244: please add references at the end of the sentence "We estimated muscle-tendon lengthening velocity by computing the numerical derivative of the muscle-tendon length data with respect to time."

Answer) According to your suggestion, we revised the sentence.

  1. Line 250: please write out SPM.

Answer) As you suggest, the entire name of SPM was written in the manuscript

  1. Lines 246 and 265: please note the publishing year for each software.

Answer) As you suggest, it was revised.

  1. Thompson, N.S.; Baker, R.J.; Cosgrove, A.P.; Corry, I.S.; Graham, H.K. Musculoskeletal modelling in determining the effect of botulinum toxin on the hamstrings of patients with crouch gait. Dev Med Child Neurol 1998, 40, 622-625.
  2. Corry, I.S.; Cosgrove, A.P.; Duffy, C.M.; Taylor, T.C.; Graham, H.K. Botulinum toxin A in hamstring spasticity. Gait Posture 1999, 10, 206-210.
  3. Papadonikolakis, A.S.; Vekris, M.D.; Korompilias, A.V.; Kostas, J.P.; Ristanis, S.E.; Soucacos, P.N. Botulinum A toxin for treatment of lower limb spasticity in cerebral palsy: gait analysis in 49 patients. Acta Orthop Scand 2003, 74, 749-755, doi:10.1080/00016470310018315.
  4. Cohen, J. Statistical power analysis for the behavioral sciences, 2nd ed ed.; Lawrence Erlbaum Associates: New York, 1988.
  5. Faul, F.; Erdfelder, E.; Lang, A.G.; Buchner, A. G*Power 3: a flexible statistical power analysis program for the social, behavioral, and biomedical sciences. Behav Res Methods 2007, 39, 175-191, doi:10.3758/bf03193146.
  6. Kang, H. Sample size determination for repeated measures design using G Power software. Anesthesia and Pain Medicine 2015, 10, 6-15, doi:10.17085/apm.2015.10.1.6.
  7. Russman, B.S.; Tilton, A.; Gormley, M.E., Jr. Cerebral palsy: a rational approach to a treatment protocol, and the role of botulinum toxin in treatment. Muscle Nerve Suppl 1997, 6, S181-193.
  8. Blumetti, F.C.; Belloti, J.C.; Tamaoki, M.J.; Pinto, J.A. Botulinum toxin type A in the treatment of lower limb spasticity in children with cerebral palsy. Cochrane Database Syst Rev 2019, 10, CD001408, doi:10.1002/14651858.CD001408.pub2.
  9. Schutte, L.M.; Hayden, S.W.; Gage, J.R. Lengths of hamstrings and psoas muscles during crouch gait: effects of femoral anteversion. Journal of orthopaedic research 1997, 15, 615-621.
  10. Arnold, A.S.; Blemker, S.S.; Delp, S.L. Evaluation of a Deformable Musculoskeletal Model for Estimating Muscle–Tendon Lengths During Crouch Gait. Annals of Biomedical Engineering 2001, 29, 263-274, doi:10.1114/1.1355277.

Reviewer 4 Report

the lack a control group is a limitation for the study

Author Response

Answer) We totally agreed with your opinion. There are some reasons for not having control group. Based on our previous studies, the parents or child did not visit to clinic for simple check up without   any financial merit including any free expensive intervention. In addition, computerized gait analysis cost a fortune in our country and thus we could not cover the cost of computerized gait analysis for the control group with our limited budget for this study. Therefore, we have to set this study as before and after design with many clinical and objective measures. We’d like to investigate the changes of the injection on kinematic and muscle length after the injection Please understand our situation.

Round 2

Reviewer 3 Report

After revision this manuscript was greatly improved. Most of the suggestions are now found in the ms.

Minor

However: in the Introduction the authors should include some sentences found in the reply to my suggestions, not only adding the ref 5-7. Also the results of refe 8-13 should be explained in full. Then the results reported in the present ms would be clearer and the reader could see what is really new.

Author Response

Detailed Response to Reviewers’ Comments

<Reviewer>

in the Introduction the authors should include some sentences found in the reply to my suggestions, not only adding the ref 5-7. Also the results of refe 8-13 should be explained in full. Then the results reported in the present ms would be clearer and the reader could see what is really new.

Answer) The references of 8 to 10 are about the best practical guidelines of botulinum toxin injection or systematic review in terms of effect. In those references, Botulinum toxin injection was considered as safe and effective intervention to control spasticity in the lower limb in people with CP. We revised the manuscript as you suggest (line 43-48). In addition, results of previous studies of ref 11-13 (ref 12-14 of 2nd revision version) were mentioned in the following sentences (line 52-56).